# Differential Expression and Clinical Relevance of C-X-C Motif Chemokine Receptor 4 (CXCR4) in Renal Cell Carcinomas, Benign Renal Tumors, and Metastases

**DOI:** 10.3390/ijms24065227

**Published:** 2023-03-09

**Authors:** Moritz Maas, Aymone Kurcz, Jörg Hennenlotter, Marcus Scharpf, Falko Fend, Simon Walz, Viktoria Stühler, Tilman Todenhöfer, Arnulf Stenzl, Jens Bedke, Steffen Rausch

**Affiliations:** 1Department of Urology, University Hospital Tübingen, Hoppe-Seyler Strasse 3, D-72076 Tübingen, Germany; 2Department of Pathology, University Hospital Tübingen, Hoppe-Seyler Strasse 3, D-72076 Tübingen, Germany; 3Clinical Trial Unit Studienpraxis Urologie, D-72622 Nürtingen, Germany; 4Medical School, Eberhard-Karls-University, D-72076 Tübingen, Germany

**Keywords:** CXCR4, renal cancer, molecular biomarker, chemokines

## Abstract

C-X-C Motif Chemokine Receptor 4 (CXCR4) is part of the human chemokine system and involved in progression and metastasis in renal cell carcinoma (RCC). However, the role of CXCR4 protein expression in RCC remains controversial. In particular, data regarding the subcellular distribution of CXCR4 in RCC and RCC metastasis as well as CXCR4 expression in renal tumors of variant histology are limited. The aim of the present study was the evaluation of the differential CXCR4 expression in RCC primary tumor and metastatic tissue as well as in variant renal histologies. In addition, the prognostic capacity of CXCR4 expression in organ-confined clear cell RCC (ccRCC) was evaluated. Three independent renal tumor cohorts (primary ccRCC cohort n_1_ = 64; cohort of various histological entities n_2_ = 146; metastatic RCC tissue cohort n_3_ = 92) were evaluated using tissue microarrays (TMA). After immunohistochemical staining for CXCR4, nuclear and cytoplasmic expression patterns were evaluated. CXCR4 expression was correlated with validated pathologic prognosticators, clinical data, and overall and cancer-specific survival. Positive cytoplasmic staining was observed in 98% of the benign and 38.9% of the malignant samples. Nuclear staining was positive for 94.1% of the benign samples and 83% of the malignant samples. The median cytoplasmic expression score was found to be higher in benign tissue than in ccRCC (130.00 vs. 0.00); median nuclear expression score analysis indicated the opposite (56.0 vs. 71.0). Within malignant subtypes, the highest expression score was seen in papillary renal cell carcinomas (cytoplasmic: 117.50, nuclear: 41.50). Within benign renal tumors, high cytoplasmic and nuclear CXCR4 expression scores were seen for oncocytomas (cytoplasmic: 100.00, nuclear: 31.00). Expression scores in RCC metastasis ranked between benign renal tissue and ccRCC in cytoplasmic and nuclear expression. Cytoplasmic CXCR4 expression was identified as a prognostic factor for OS and CSS (*p* = 0.042; *p* = 0.019). Multivariate analysis including clinicopathological parameters did not reveal an independent prognostic character of CXCR4 expression. CXCR4 expression differs significantly within benign lesions and renal neoplasms. Cytoplasmic and nuclear expression of CXCR4 could be detected across all RCC subtypes. The prognostic value of CXCR4 in ccRCC was confirmed in univariate analysis.

## 1. Introduction

Renal cell carcinoma (RCC) represents 3% of all tumors worldwide and remains one of the most lethal urological malignancies, despite recent advances in diagnostics and systemic therapies [1]. C-X-C Motif Chemokine Receptor 4 (CXCR4) is a membrane-bound G-protein coupled receptor in the chemokine axis. This complex system of 48 chemokines and 23 receptors plays a crucial role in various signaling pathways. The most outstanding role is within the immune system, while other functions include hematopoiesis, embryogenesis, and involvement in inflammatory processes and neovascularization [2,3,4]. Beyond that, the chemokine system and particularly CXCR4 with its ligand CXCL12 are considered to be of importance in various solid tumor types [5]. The CXCR4/CXCL12 axis appears to be tumor-promoting; increased expression of CXCR4 was linked to more aggressive behavior and metastasis in various tumor types, including prostate cancer [5,6,7,8]. Similarly, meta-analyses demonstrated shorter overall and progression-free survival (OS/PFS) also for clear-cell (cc)RCC patients with high CXCR4 expression [9,10]. CXCR4 has consequently been considered as a prognostic marker in RCC. However, current guidelines urge additional validation before incorporation into routine clinical practice [11].

CXCR4 protein expression was reported to be higher in low Fuhrmann grades, while there are no discernible variations between pathological stage and CXCR4 expression [10]. Notably, expression in metastatic tissue was reported to be significantly higher than in non-metastatic primary RCC tissue [12,13]. Additionally, there is an imbalance in the analysis of CXCR4 expression within various renal tissues: while a large body of evidence is available for clear cell renal cell carcinoma (ccRCC), there is a lack of information on CXCR4 expression in benign renal tissue, including oncocytomas and angiomyolipomas, as well as the differential expression of CXCR4 in ccRCC in comparison to other malignant subtypes. In addition, only a few studies have addressed the comparative analysis of CXCR4 expression between metastatic tissue and primary RCC [12,13].

Increasing attention is further given to a more detailed evaluation of CXCR4′s subcellular localization using immunohistochemistry (IHC). CXCR4 contributes to tumor progression and metastasis of solid tumors through two distinct pathways that are both G protein-dependent (AKT, PI3K, mTor, EGFR, among others) and -independent (JAK/STAT, p53, MAPK, ERK, among others) [14,15,16,17]. Inhibitors of the CXCR4–CXCL12 axis have been developed as a therapeutic approach considering the prognostic relevance of CXCR4 overexpression and the role of this axis in tumor growth (AMD3100/Plerixafor, Nox-A12 and others) [14,18,19]. A randomized phase 2 trial comparing the effectiveness of one of these CXCR4 inhibitors (LY2510924) in combination with sunitinib vs. sunitinib alone was conducted for the treatment of metastatic RCC, but failed to demonstrate an improvement in OS or PFS [20]. In other tumor entities, the results of phase I (breast cancer, NCT 01837095) and phase II studies (pancreatic cancer, NCT 04177810) are still pending, but at present, CXCR4 inhibitors have not met the expectations of clinical efficacy [14]. A growing number of studies suggest that this could be explained by the presence of an intracellular localized CXCR4 protein that remains unattainable for antibodies targeting predominantly the cell surface receptor [21,22,23]. For the subcellular distribution of CXCR4 in renal malignancies, heterogeneous data with regard to, e.g., membranous or cytoplasmic expression have been reported [12,24,25].

The objective of the current study was a comprehensive analysis of CXCR4 expression in primary renal cell carcinoma, metastatic tissue, and other histological subtypes of renal neoplasms including benign lesions. Special attention was given to the subcellular distribution of CXCR4. Furthermore, we evaluated the prognostic character of CXCR4 in organ-confined ccRCC.

## 2. Results

The n_1_ cohort included 64 patients with ccRCC, in which 27 (42.2%) were women and 37 (57.8%) were men. The median age at surgery was 64.2 years (35.7–88 years), and the median follow-up was 53.34 months. Table 1 shows the pathological characteristics of the n_1_ cohort.

The n_2_ cohort consisted of 146 patients with renal tumors of various histological entities, of which 87 were ccRCCs (59.59%), 14 papillary RCCs (9.59%), 7 chromophobic RCCs (4.79%), 16 oncocytomas (10.96%), 10 angiomyolipomas (6.85%), 7 cysts (4.79%), and 5 of unknown histology (3.43%). Median age at surgery was 64.08 years (24.30–89.82 years) (Table 2). The n_3_ cohort consisted of 92 patients with metastatic RCC. Metastasis was synchronous in 27 patients (29.35%), metachronous in 62 patients (67.39%). For 3 patients (3.26%), time of metastasis could not be determined. The histology of the primary tumor was ccRCC in 83 patients (90.22%), followed by papillary RCC (3.26%) and chromophobic RCC (2.17%) with 3 and 2 patients, respectively. Metastases most frequently were located in the lung (35 cases, 25.55%), followed by soft tissues (21 cases; 15.33%), lymph nodes (15 cases, 10.95%), and adrenal gland (11 cases; 8.03%) (Table 3).

### 2.1. Expression of CXCR4

Cytoplasmic expression of CXCR4 could be evaluated in 118 samples of n_1_ and 125 samples of n_2_. Nuclear expression was assessed in 113 samples of n_1_ and 121 samples of n_2._ Analysis of cytoplasmic and nuclear expression in cohort n_3_ separated according to metastasis site was obtained for 63 samples each. Pooled analysis for histologically confirmed ccRCC metastases was feasible in 96 (cytoplasmic) and 95 samples (nuclear), respectively (Appendix A).

In the n_1_ cohort, cytoplasmic CXCR4 expression was observed in 98% of benign kidney tissue samples, whereas only 38.9% of the ccRCC tumor tissue samples showed an expression of CXCR4. Median cytoplasmic staining scores differed significantly between benign and tumor tissue (*p* < 0.0001). Nuclear CXCR4 expression was present in 94.1% of benign tissue samples and 83% of ccRCC. Median nuclear staining score in benign tissue samples was 56.0 (95% CI 42.00–60.00) and 71.0 (95% CI 59.63–78.00) for tumor tissue samples (*p* = 0.0124). Figure 1 illustrates the staining patterns for the patient cohort n_1_.

Analysis of nuclear and cytoplasmic expression in cohort n_2_ revealed heterogeneous results: the cytoplasmic expression score was highest in papillary RCCs (median: 117.50; 95% CI 5.00–245.50), followed by oncocytomas (100.0; 95% CI 24.54–173.51). CcRCCs showed very low staining scores, likewise chromophobic RCCs. The median cytoplasmic staining scores of papillary RCC differed significantly from all other entities except oncocytoma, whereas the expression in ccRCC and chromophobe RCC was only significantly different from papillary RCC and oncocytoma (*p* < 0.05, each).

The highest scores for nuclear expression were evident in cysts (median: 90), followed by papillary RCCs (41.50, 95% CI 16.00–92.27) and oncocytomas (31.00, 95% CI 17.79–68.09).

Post hoc analysis showed significant differences in staining scores of chromophobic RCC, between papillary RCC and oncocytoma (*p* < 0.05). The comparison of ccRCC with papillary RCC, chromophobic RCC, oncocytoma and angiomyolipoma revealed no significant difference. Figure 2 illustrates the staining patterns for n_2_ and shows the post hoc analysis between the different entities.

The comparison of cytoplasmic and nuclear CXCR4 expression in cohort n_3_ revealed no significant differences depending on metastasis location. High expression scores were observed for cytoplasmic expression in pleura (190) and skin (185). For local recurrence, high CXCR4 expression values were shown for both cytoplasmic (300) and nuclear counts (86). Figure 3 illustrates the staining patterns for cohort n_3_.

Finally, the comparison of CXCR4 staining between benign tissue, ccRCC, and metastatic tissue of ccRCC revealed significant differences in expression scores for both cytoplasmic and nuclear expression. Here, benign tissue showed the highest cytoplasmic scores, as compared to primary RCC and RCC metastasis. For cytoplasmic staining, a significant difference was evident in each subgroup comparison (*p* < 0.05, each). Nuclear expression showed the highest staining scores for primary RCC (71.00; 95% CI 59.63–78.00), followed by metastatic tissue (60.00; 95% CI 44.51–75.74) and benign tissue (56.00; 95% CI 42.00–60.00). Nuclear expression in primary RCC differed significantly from benign tissue, and the difference between benign tissue and metastatic tissue was likewise significant (*p* < 0.05, each). No significant difference in nuclear staining between primary RCC and metastases were noted (Figure 4). Regression analysis revealed no significant correlation of cytoplasmic and nuclear staining scores (*p* = 0.47).

### 2.2. CXCR4 Expression in Correlation with Clinicopathological Data in Cohort n_1_

Correlation analysis of CXCR4 expression with clinicopathologic parameters demonstrated a significant positive correlation of cytoplasmic expression with lymphovascular invasion. The remaining clinicopathologic parameters (T stage > T2, positive lymph node status, vascular invasion, grading > G2, sarcomatoid differentiation, and necrosis) did not yield a statically significant correlation with nuclear or cytoplasmic CXCR4 expression. Table 4 summarizes differential CXCR4 expression in correlation with clinicopathological variables.

### 2.3. Survival Analysis

Survival analysis was performed for cohort n_1_. The mean OS for the entire cohort n_1_ was 71.73 months (95% CI 64.33–79.12 months); patients with cytoplasmic expression of CXCR4 showed a significantly decreased OS of 58.8 months compared to patients without cytoplasmic expression (74.8 months, *p* = 0.042). A significant difference was also evident in CSS in correlation with cytoplasmic expression. The mean CSS was 61.44 months (95% CI 48.67–74.21) in the presence of cytoplasmic CXCR4 expression, and 78.69 months without (95% CI 71.69–84-81), respectively (*p* = 0.019). Kaplan–Meier survival analysis for OS and CSS in dependence of cytoplasmic expression is illustrated in Figure 5. Nuclear expression of CXCR4 failed to show a statistically significant association with OS and CSS (Appendix A).

In a multivariate Cox regression analysis model, after the adjustment for relevant prognostic clinicopathological parameters in ccRCC (T-stage ≥ T2, sarcomatoid features, positive lymph node status, vascular invasion, grading > G2), an independent prognostic character of CXCR4 expression was not confirmed (Table 5).

## 3. Discussion

Our study showed that CXCR4 expression is highly variable both between the evaluated histological subtypes and between the analyzed subcellular compartments. Absolute staining scores showed low values for cytoplasmic CXCR4 expression in both n_1_ and n_2_. While cytoplasmic expression between ccRCC and benign kidney tissue within cohort n_1_ revealed a significant difference, this was not confirmed in nuclear analysis.

Owing to the different applied scores, cytoplasmic and nuclear staining scores may not be directly compared, but it was noticeable that within the n_2_ cohort, cytoplasmic staining scores of ccRCC were at the lower end of the scale, whereas for nuclear staining they ranged in the middle.

Papillary RCCs, on the other hand, showed a significantly high cytoplasmic expression score compared to the other RCC subtypes and likewise exhibited a reasonably high expression level for nuclear staining, although the significance level was reached only in comparison to chromophobic RCC. The chromophobic RCCs, by contrast, showed low expression levels for both cytoplasmic and nuclear evaluation. Therefore, clearly distinct expression patterns could be attested here in the comparison of the different renal tumor types.

There was no such clear trend in the comparison of benign tissue with malignant tissue: Benign tissue samples from n_1_ showed both significantly higher frequency of staining and stronger staining intensity for cytoplasmic staining. Integrating our results into the existing evidence was challenging, as the CXCR4 expression showed a heterogeneous picture in benign kidney tissue. Wang et al. described a higher cytoplasmic CXCR4 expression in malignant tissue (predominantly ccRCCs) compared to benign tissue, which was likewise observed by Li et al. in malignant tissue from ccRCCs [12,24]. Our observation of rather high staining intensity in benign tissue might have been related to the staining property of tubules, for which intense expression of CXCR4 has been described previously [24,26,27].

However, a clear pattern regarding consistently high cytoplasmic expression in benign tissue and consistently low cytoplasmic expression in malignant tissue or vice versa could not be established within our analysis. Additionally in the analysis of benign kidney tissue, we evaluated expression within benign kidney lesions of various histological geneses: Oncocytomas as benign tumors of epithelial origin showed significantly higher cytoplasmic CXCR4 expression than ccRCCs, whereas such a difference could not be shown for angiomyolipomas as tumors of mesenchymal origin. Although direct comparability was limited, a trend could be noted regarding the change in staining intensity between cytoplasmic staining (malignant < benign) and nuclear staining (malignant > benign). In the comparison of benign kidney tissue, ccRCCs, and metastases of ccRCCs, both cytoplasmic and nuclear CXCR4 expression in primary ccRCC ranked between benign tissue and metastatic tissue (Figure 4). Analysis of a cohort of samples from different metastatic sites (n_3)_ demonstrated that a relevant correlation between metastasis localization and expression behavior did not exist.

CXCR4 is a G protein-coupled receptor with expected main localization in the membrane [28]. In various tumor entities, however, an intracellular protein localization of CXCR4 has been described [21,29]. This is considered to be of functional importance, particularly in light of the limited results of CXCR4 antagonists in therapeutic trials so far [14,20]. A simple ligand-based approach does not seem to be sufficient for the prevention of the pro-tumorigenic effect of CXCR4 and ligand-independent, transcriptional and post-transcriptional effects seem to have more impact than previously thought [14].

Our study confirms that cytoplasmic and nuclear expression of CXCR4 is detectable in ccRCC and may contribute to the reduced efficacy of the ligand-based therapeutic approach. Intracellular CXCR4 expression was also observed in other studies: While D’Alterio et al. described the main staining as membranous, various other authors such as Wehler et al., Wang et al., and Li et al. found a primarily cytoplasmic staining [12,24,25,27]. Predominant cytoplasmic CXCR4 expression could be caused by internalization of membrane-bound receptors. It is known that continuous activation of CXCR4 by its ligand CXCL12 leads to internalization of the receptor [15]. High CXCL12 levels were detected in metastases as well as intratumorally [30,31]. Therefore, cytoplasmic expression may be an indirect sign of a high rate of CXCL12/CXCR4 interaction.

Regarding nuclear CXCR4 staining, most authors described a reduced expression compared to cytoplasmic staining in renal malignancies [12,24,25,27,31]. Remarkably, a change in staining patterns between benign kidney tissue and malignant tissue for cytoplasmic and nuclear staining was observed in our study. Translocation of CXCR4 to the nucleus could hypothetically be a reason for this transformation. Nuclear expression of CXCR4 has been described in several solid tumor types including colorectal cancer, breast cancer, and prostate cancer [22,32,33]. In addition, in prostate cancer, a mechanism has been described as leading to a translocation of CXCR4 to the nucleus, where it remains functional, ultimately resulting in increased metastatic potential and limited prognosis [22]. For RCC, Bao et al. described that nuclear localization of CXCR4 contributed to poor prognosis as well as development of metastases. They showed that CXCR4 interacted with HIF1-alpha and, after this interaction, translocated to the nucleus, where the downstream genes of HIF1-alpha were increasingly being transcribed [34].

The results of cytoplasmic and nuclear expression analysis in the other RCC subtypes in our study were consistent with the literature: Rasti et al. and Floranovic et al. described higher CXCR4 expression in papillary renal cell carcinomas compared to ccRCC [26,35]. On the other hand, they reported more pronounced nuclear expression in chromophobic RCC as compared to ccRCC [35]. Comparison of CXCR4 expression in metastatic tissue with primary tumor expression revealed a significantly higher IR score for cytoplasmic expression in metastatic tissue, whereas nuclear expression was significantly stronger in primary tumor tissue. The body of studies regarding the evaluation of metastases is notably limited. Wang et al. described a higher CXCR4 expression in metastatic tissue than in primary tumor tissue, both showing higher expression than benign tissue. For primary RCC, CXCR4 expression was predominantly observed in the cytoplasmic membrane, while metastatic tissue showed a predominantly cytoplasmic-only staining pattern. Differences with our results are apparent here, although we also observed a higher cytoplasmic expression score in metastatic tissue than in primary RCC. The sample numbers included by Wang et al. were 43 primary ccRCCs and 28 metastases (21 lymph node metastases, 4 adrenal gland metastases, 2 bone metastases, 1 brain metastasis) [12]. Zhao et al. evaluated the CXCR4 expression in primary RCC compared to bone metastases. They describe a weak expression of CXCR4 at the cell surface and the cytoplasm near the membrane in primary RCC, whereas bone metastases showed an expression increase. A more detailed description of subcellular expression patterns was not reported [13].

An intriguing observation was the high cytoplasmic expression of CXCR4 in oncocytomas. In a feasibility study, Werner et al. evaluated CXCR4-guided functional imaging in solid tumors [36], indicating the putative perspective of CXCR4-guided imaging. In light of the sometimes challenging distinction between oncocytomas and ccRCCs in conventional imaging, an indication for this type of functional imaging may emerge given the high CXCR4 expression differences.

For a clinically well defined cohort of primary ccRCC (n_1_), we further correlated the CXCR4 expression with prognostically relevant clinicopathological parameters and survival. A positive correlation was shown exclusively for lymphovascular invasion, confirming the independent character of CXCR4 and indicating its potential as a biomarker, as suggested by recent guidelines [11]. A positive correlation with microvascular invasion was also described by Rasti et al. [35]. However, the association with further parameters of advanced tumors described in other studies could not be confirmed within our cohort n_1_ [27,37].

In Kaplan–Meier analyses, patients with cytoplasmic CXCR4 expression showed significantly longer OS and CSS. Multivariate analysis including established prognostic markers (high tumor stage, positive lymph node status, venous invasion, lymphovascular invasion, sarcomatoid differentiation, and grading > G2) did not confirm CXCR4 as an independent prognosticator. It should be highlighted in this context that within this multivariate analysis, the aforementioned clinically validated and established prognostic factors likewise lost their statistical significance from the univariate analysis. Although some factors can be considered as not completely independent from the others (e.g., Fuhrman grading and T-stage), this observation must be considered as a limitation of our work and may be an indicator of an insufficient number of cases in our cohort n_1_. This limitation was confirmed when analyzing other studies, which clearly indicated that prognostic significance can be assumed for CXCR4 expression: Both high protein expression of CXCR4 [24,25,26,31,35,38,39] and high mRNA expression [40,41] of CXCR4 have negative prognostic significance for survival.

The present study is further limited by its retrospective approach and the sample size. This is particularly notable for the analysis of the n_3_ metastatic cohort, in which only solitary samples were available for certain localizations. Due to the retrospective character of the study, existing pathological and radiological findings were used for the collection of clinicopathological data, which led to the lack of an ISUP/WHO classification for renal cell carcinoma. Moreover, analyses of central nervous system metastases could not be given due to the unavailability of corresponding specimens. Comparison of primary tumor expression with corresponding metastatic tissue from the same patient was not possible due to the independent character of our patient cohorts. Intraindividual comparison of CXCR4 expression between primary tumor and metastasis would further increase data value; a potential approach would be the design and evaluation of larger prospective tissue biobanks. The study was further limited by the inherent limitations of semi-quantitative TMA analysis, such as inter-observer variability and variable performance of immunoassays.

In conclusion, our study did not show a clear pattern of CXCR4 expression when comparing benign tissues, primary RCCs of different histologic subtypes and metastatic tissues. Instead, the immunohistochemical expression of this receptor showed a highly variable pattern. The distinct cytoplasmic and nuclear expression of this expected membrane protein is remarkable, especially in the context of the so far disappointing results of CXCR4 inhibitors in clinical trials.

Furthermore, a prognostic value of positive cytoplasmic CXCR4 expression is emerging in clear cell renal cell carcinoma. Our multivariate analyses could not confirm an independent character of this positive cytoplasmic CXCR4 expression. However, also-validated clinical prognostic factors could not prove their independence in this analysis, suggesting that future analysis of larger cohorts may be able to validate CXCR4 as an independent prognostic factor.

## 4. Materials and Methods

### 4.1. Patient Cohorts

The expression of CXCR4 was evaluated in three independent institutional cohorts (primary ccRCC tumor cohort n_1_ = 64 patients, cohort of various histological entities n_2_ = 146, and a metastatic tissue cohort of n_3_ = 92 patients with tissue from, in total, 137 RCC metastases) using tissue microarrays (TMA) from formalin-fixed, paraffin-embedded tissue samples (see Table 1, Table 2 and Table 3 for patient characteristics). Patients underwent surgery between 2007–2010 (n_1_), 2012–2013 (n_2_) and 1990–2012 (n_3_). No individual patient was included in more than one cohort. Tumors were pathologically classified according to the UICC TNM classification 2002 [42].

Expression analysis within cohort n_2_ was performed by histological entities separately. For the expression analysis according to metastatic site in cohort n_3_, 63 reliably localizable samples without restriction for primary RCC type were analyzed; only one sample was permitted per patient. For comparison of benign kidney tissue, ccRCCs and metastatic tissue, representative ccRCC samples from cohort n_1_ with corresponding benign kidney tissue and histologically verified ccRCC metastases from cohort n_3_ were evaluated; multiple ccRCC metastatic tissue samples were permitted per patient from cohort n_3_.

Patient characteristics for cohort n_1_ and n_2_, were collected using the hospital’s internal information system. Follow-up data of cohort n_1_ were collected by contacting office-based physicians, or patients directly. For the comparison of ccRCC tissue, benign kidney tissue and metastatic tissue, ccRCC samples and benign tissue samples of cohort n_1_ were combined with histologically confirmed ccRCC metastases from cohort n_3_. Evaluated clinicopathological parameters for cohort n_1_ and n_2_ included validated pathological prognostic factors (T-stage; lymph node status (N-stage); vascular invasion (V-Status); lymphovascular invasion (L-Status), grading ≥ G2, necrosis, sarcomatoid features). All participants gave written informed consent. The study was approved by the Tübingen University Ethics Committee (078/2012B02).

### 4.2. TMA and Immunohistochemistry

TMA slides were created using a tissue arrayer (Beecher Instruments, Silver Springs, MD, USA) after histological evaluation of hematoxylin and eosin-stained slides as previously described [43]. Representative tumor areas were identified microscopically, and cores of 0.6mm diameter were collected from two locations within this area to represent intratumoral heterogeneity (applicable to ccRCC in cohort n_1_, malignant and benign renal tumors in cohort n_2_, and metastatic tissue in cohort n_3_). For the benign comparative samples, areas of normal kidney tissue were identified microscopically, and one core was obtained here. For immunohistochemistry, sections were first deparaffinized and rehydrated. Heat-induced antigen retrieval was applied by usage of a microwave oven. Endogenous peroxidase activity was blocked by incubation in peroxide block with 3% H_2_O_2_ solution. Unspecific background staining was minimized by incubation with a protein blocking solution (Bovine serum albumin (BSA) blocking buffer, Sigma Aldrich, Darmstadt, Germany).

TMA slides were then incubated with the primary antibody (recombinant rabbit monoclonal anti-CXCR4 antibody, UMB 2, abcam plc, Cambridge, UK) diluted 1:100 using a commercial diluent (Cat. No.: ZUC025-100, Zytomed Systems GmbH, Berlin, Germany) at 4 °C overnight. For visualization, a commercially available DAB detection kit (Cat. No: POLHRP-100, Zytomed Systems GmbH, Berlin, Germany) was applied: After washing, the enhancement reagent was applied and incubated. Following a second washing step, horseradish peroxidase (HRP)-polymer was applied. Addition of chromogenic substrate DAB to the enzymatic peroxidase reaction led to a quantitative concentration-equivalent color precipitation (dark brown reaction product) in locations where the primary antibody had bound to the tissue, identified by light microscopy. Counterstaining was performed by using Papanicolaou staining solvent (Art. No.: 109253/109254, Sigma-Aldrich, Darmstadt, Germany), leading to a light blue appearance of the cell borders and nuclei for staining interpretation. Negative controls were processed by omitting the primary antibody.

### 4.3. Semi-Quantitative Evaluation of TMAs

TMA evaluation was performed in a blinded fashion by two independent reviewers (SR, AK). After negative membranous pilot staining tests, CXCR4 was evaluated for cytoplasmic and nuclear staining patterns in a randomly selected, representative reference area. Divergent results were reevaluated. For cytoplasmic staining, an immunohistochemistry reactivity score (IRS) was generated by multiplication of the staining intensity (0: no staining–3: intensive staining) with the proportion (0–100%) of the stained cells, resulting in an IRS range from 0–300. For nuclear staining, the count of stained cell nuclei/100 cells in representative area was evaluated and transferred into a continuous score. Figure 6 illustrates representative staining intensities for CXCR4 staining expression.

### 4.4. Statistical Analysis

Patterns of CXCR4 expression were assessed by applying the Mann–Whitney *U* test for comparison of two groups and the Kruskal–Wallis test for comparison of more than two groups. Correlation of CXCR4 expression and clinicopathological parameters was performed using the Chi-Square test. Kaplan–Meier analyses were performed to evaluate overall survival (OS) and cancer-specific survival (CSS). For cytoplasmic expression, groups were dichotomized according to their CXCR4 expression (positive: staining intensity ≥ 1, or negative: staining intensity = 0). For nuclear expression, groups were dichotomized into positive expression (detectable nuclear staining of any intensity) and negative expression (no detectable staining). Differences between subgroups were evaluated using Log-rank test. Univariate and multivariate Cox proportional regression analyses were performed to assess the correlation of clinical parameters with survival. Statistical significance was regarded as *p* < 0.05. Statistical analysis was performed using commercial software (MedCalc Version 12.5, Ostend, Belgium).

## 5. Conclusions

CXCR4 is a receptor of the chemokine system and has pro-tumorigenic characteristics. Overexpression of CXCR4 has been described in RCC with negative prognostic significance for OS and CSS. In contrast to protein expression data for CXCR4 in ccRCC, limited evidence exists for benign renal tissue, papillary and chromophobe RCC, benign tumors and metastatic tissue. In this study, a comprehensive analysis of CXCR4 expression in these tissue types demonstrated that CXCR4 is detectable subcellularly in both cytoplasm and nucleus. CXCR4 expression differed significantly in the different malignant subtypes, and there was a transition in differential expression between cytoplasm and nucleus when comparing benign renal tissue and ccRCC. A consistent increment of CXCR4 expression from benign tissue to primary RCC to metastatic tissue was not evident. Moreover, high cytoplasmic CXCR4 expression appeared as a characteristic in oncocytomas, which might be clinically meaningful in differential histopathology or functional imaging in the future. Moreover, the prognostic significance of CXCR4 in ccRCC could be confirmed.

## Figures and Tables

**Figure 1 ijms-24-05227-f001:**
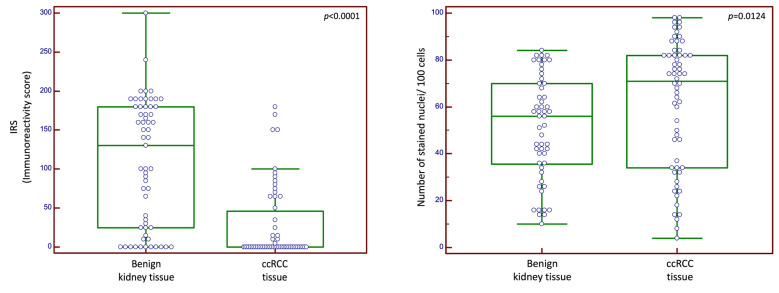
Cytoplasmic and nuclear expression scores for cohort n_1_ (primary ccRCC cohort). Abbreviations: IRS: immunoreactivity score; ccRCC: clear cell renal cell carcinoma. Green box represents median and 1st and 3rd quartile. Purple circles are individual values, circles outside the box are outside values.

**Figure 2 ijms-24-05227-f002:**
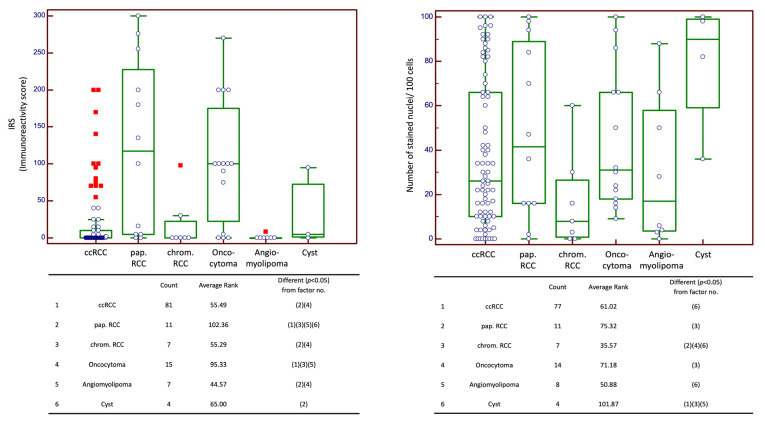
Cytoplasmic and nuclear expression scores for cohort n_2_ (cohort of various histological entities) with Kruskal–Wallis and post hoc analysis. Abbreviations: IRS: immunoreactivity score; ccRCC: clear cell renal cell carcinoma; pap. RCC: papillary renal cell carcinoma; chrom. RCC: chromophobic renal cell carcinoma. Green box represents median and 1st and 3rd quartile. Purple circles are individual values, circles outside the box are outside values. Red squares are far out values.

**Figure 3 ijms-24-05227-f003:**
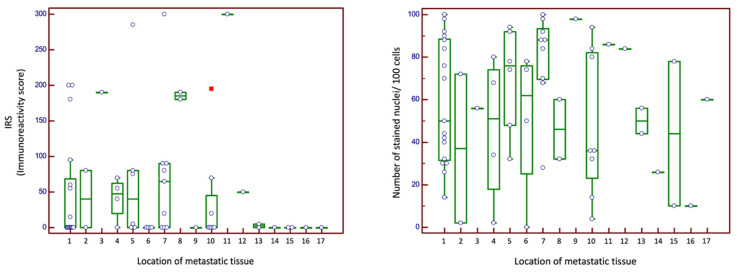
Cytoplasmic and nuclear expression scores for cohort n_3_ (metastatic tissue cohort). 1 lung; 2 liver; 3 pleura; 4 adrenal gland; 5 lymph node; 6 pancreas; 7 soft tissue; 8 skin; 9 larynx; 10 bone; 11 local recurrence; 12 sympathetic trunk; 13 diaphragm; 14 mesentery; 15 small intestine; 16 spleen; 17 peritoneum. Abbreviations: IRS: immunoreactivity score. Green box represents median and 1st and 3rd quartile. Purple circles are individual values, circles outside the box are outside values. Red squares are far out values.

**Figure 4 ijms-24-05227-f004:**
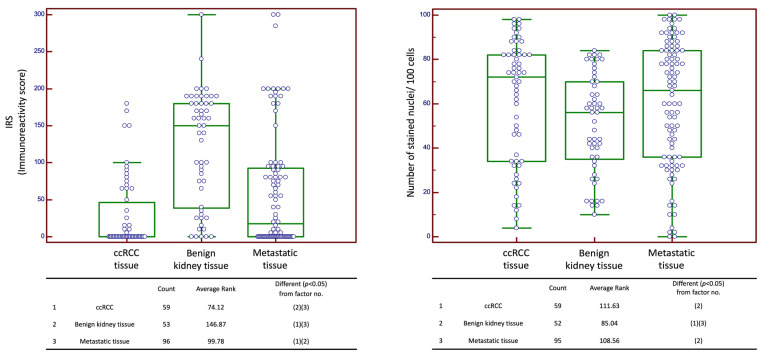
Cytoplasmic and nuclear expression scores for combined cohort of n_1_ and n_3_. The evaluated samples represent ccRCC samples with their corresponding benign tissue from cohort n_1_ and histologically confirmed ccRCC metastases from cohort n_3_. Abbreviations: IRS: immunoreactivity score; ccRCC: clear cell renal cell carcinoma. Green box represents median and 1st and 3rd quartile. Purple circles are individual values, circles outside the box are outside values.

**Figure 5 ijms-24-05227-f005:**
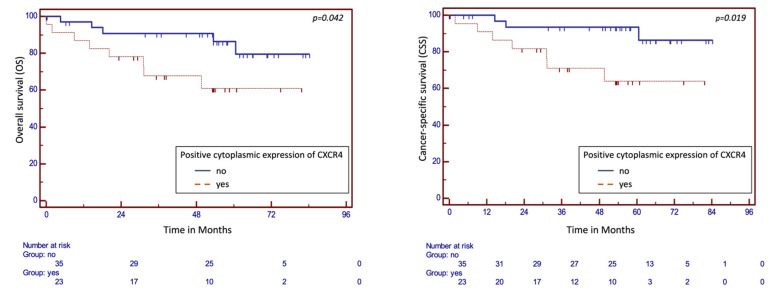
Kaplan–Meier analysis for cohort n_1_: Overall survival (OS) in dependence of positive cytoplasmic CXCR4 expression. Cancer-specific survival (CSS) in dependence of positive cytoplasmic CXCR4 expression.

**Figure 6 ijms-24-05227-f006:**
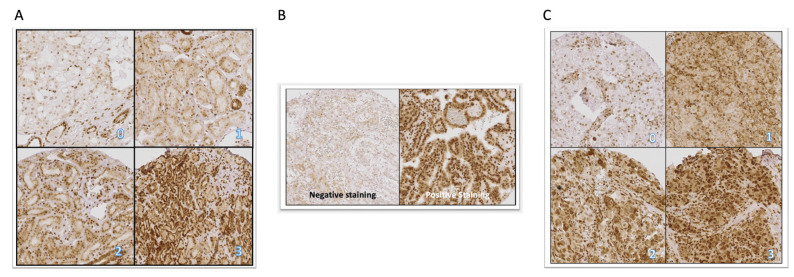
Exemplary illustration of the staining intensities. (**A**) Staining intensities 0–3 in cytoplasmic staining for CXCR4 in malignant samples. Immunoreactivity score (IRS) was obtained by multiplication of the intensity with the percentage (0–100%) of the stained cells, resulting in scores in the range of 0–300. (**B**) Illustration of negative and positive nuclear staining for CXCR4 in malignant samples. (**C**) Staining intensities 0–3 in cytoplasmic staining for CXCR4 in benign kidney tissue. Immunoreactivity score (IRS) was obtained by multiplication of the intensity with the percentage (0–100%) of the stained cells, resulting in scores in the range of 0–300.

**Table 1 ijms-24-05227-t001:** Characteristics of patient cohort n_1_: primary tumor cohort.

	*n*	%
**Total number of patients**	64	100
Male	37	57.8
Female	27	42.2
**Age in years**	Median 64.2	Range 35.7–88.0
**Pathological stage**		
T1a	18	28.1
T1b	16	25.0
T2a	5	7.8
T2b	1	1.6
T3a	10	15.6
T3b	14	21.9
**Lymph node status**		
N0	57	89.1
N1	7	10.9
**Distant metastatic status**		
M0	54	84.4
M1	9	14.1
Mx	1	1.5
**R-Status**		
R0	60	93.8
R1	4	6.2
**L-Status**		
L0	61	95.3
L1	3	4.7
**V-Status**		
V0	46	71.9
V1	17	26.5
V2	1	1.6
**Fuhrman grading**		
G1	14	21.8%
G2	41	64.1%
G3/G4	9	14.1%
**Sarcomatoid features**		
Yes	4	6.2
No	60	93.8
**Tumor necrosis**		
Yes	9	14.1
No	55	85.9
**Primary tumor size in cm**	Median 5.5	Range 1.4–16.0
**FU time (months) from date of diagnosis of primary ccRCC**	Mean 47.02 Median 53.34	95% CI (mean) 42.03–52.02 Range (Median) 0.00–87.72

Abbreviations: FU time: Follow-up time. ccRCC: clear cell renal cell carcinoma; R-Status: residual tumor classification; R0: no residual tumor; R1: residual tumor; L-Status: lymphovascular invasion status; L0: no lymphovascular invasion; L1: lymphovascular invasion; V-Status: vascular invasion status; V0: no vascular invasion; V1: renal vein with tumor invasion; V2: vena cava with tumor invasion

**Table 2 ijms-24-05227-t002:** Characteristics of patient cohort n_2_: cohort of various histological entities.

	N_2_: Cohort of Various Histological Entities					
	** *n* **			%		
**Total number of patients**	**146**			100		
Male	90			61.64		
Female	56			38.36		
**Age in years**	Median in years 64.08			Range 24.30–89.82		
**Malignant tumors within this cohort**						
	**ccRCC**		**Pap. RCC**		**Chrom. RCC**	
	*n*	%	*n*	%	*n*	%
**Number of Patients**	87	59.59	14	9.59	7	4.79
Male	53	36.30	11	7.53	4	2.74
Female	34	23.29	3	2.05	3	2.05
**Age in years**	Median 65.08	Range 27.66–89.82	Median 63.38	Range 37.72–81.11	Median 46.54	Range 44.70–81.15
**Pathological Stage**						
T1a	42	28.77	6	4.11	2	1.37
T1b	18	12.33	3	2.05	3	2.05
T2a	4	2.74	2	1.37	2	1.37
T2b	0	0.00	0	0.00	0	0.00
T3a	20	13.70	2	1.37	0	0.00
T3b	2	1.37	1	0.68	0	0.00
T3c	1	0.68	0	0.00	0	0.00
T4	0	0.00	0	0.00	0	0.00
**Lymph node status**						
N0	79	54.11	12	8.22	7	4.79
N1	6	4.11	2	1.37	0	0.00
Nx	2	1.37	0	0.00	0	0.00
**Distant metastatic status**						
M0	74	50.68	1	0.68	7	4.79
M1	11	7.53	13	8.90	0	0.00
Mx	2	1.37	0	0.00	0	0.00
**R-Status**						
R0	77	52.74	11	7.53	7	4.79
R1	7	4.79	2	1.37	0	0.00
RX	3	2.05	1	0.68	0	0.00
**L-Status**						
L0	85	58.22	11	7.53	7	4.79
L1	2	1.37	3	2.05	0	0.00
**V-Status**						
V0	72	49.32	13	8.90	7	4.79
V1	11	7.53	0	0.00	0	0.00
V2	4	2.74	1	0.68	0	0.00
**Fuhrman grading**						
G1	25	17.12	2	1.37	1	0.68
G2	48	32.88	9	6.16	5	3.42
G3-4	13	8.90	2	1.37	1	0.68
Gx	1	0.68	1	0.68	0	0.00
**Sarcomatoid features**						
Yes	3	2.05	1	0.68	0	0.00
No	84	57.53	13	8.90	7	4.79
**Tumor Necrosis**						
Yes	21	14.38	6	4.11	1	0.68
No	66	45.21	8	5.48	6	4.11
**Primary tumor size in cm**	Median 3.5	Range 0.5–15.00	Median 3.9	Range 1.5–12.00	Median 4.3	Range 2.00–7.80
**Benign tumors within this cohort**						
	**Oncocytoma**		**Angiomyolipoma**		**Cyst**	
	*n*	*%*	*n*	*%*	*n*	*%*
**Number of patients**	16	10.96	10	6.85	7	4.79
Male	9	6.16	3	2.05	6	4.11
Female	7	4.79	7	4.79	1	0.68
**Age in years**	Median 62.56	Range 51.38–77.86	Median 60.88	Range 24.30–74.13	Median 61.28	Range 32.56–78.82
**Unknown tumors within this cohort**						
	*n*			*%*		
**Number of patients**	5			3.42		
Male	3			2.05		
Female	2			1.37		
**Age in years**	Median 60.4			Range 34.32–74.85		

Abbreviations: ccRCC: clear cell renal cell carcinoma; pap. RCC: papillary renal cell carcinoma; chrom. RCC: chromophobic renal cell carcinoma; R-Status: residual tumor classification; R0: no residual tumor; R1: residual tumor; Rx: residual tumor status unknown; L-Status: lymphovascular invasion status; L0: no lymphovascular invasion; L1: lymphovascular invasion; V-Status: vascular invasion status; V0: no vascular invasion; V1: renal vein with tumor invasion; V2: vena cava with tumor invasion

**Table 3 ijms-24-05227-t003:** Characteristics of patient cohort n_3_: metastatic tissue cohort.

N_3_: Metastases cohort		
	*n*	%
**Total number of patients with met. RCC**	92	100%
Total number of metastases	137	100%
**Histology of primary tumor**		
ccRCC	83	90.22%
Pap. RCC	3	3.26%
Chrom. RCC	2	2.17%
Unknown	4	4.35%
**Metastasis**		
metachronous	62	67.39%
synchronous	27	29.35%
unknown	3	3.26%
**Metastatic site**		
Lung	35	25.55%
Soft tissue	21	15.33%
Bone	18	13.14%
Lymph node	15	10.95%
Adrenal gland	11	8.03%
Pancreas	6	4.38%
Liver	5	3.65%
Local recurrence	4	2.92%
Diaphragm/ Muscle	3	2.19%
Pleura	2	1.46%
Testicle	2	1.46%
Small intestine	2	1.46%
Skin	1	0.73%
Larynx	1	0.73%
Sympathetic trunk	1	0.73%
Mesentery	1	0.73%
Peritoneum	1	0.73%
Spleen	1	0.73%
Brain	1	0.73%
Unknown	6	4.38%

Abbreviations: ccRCC: clear cell renal cell carcinoma; pap. RCC: papillary renal cell carcinoma; chrom. RCC: chromophobic renal cell carcinoma.

**Table 4 ijms-24-05227-t004:** Correlation of positive CXCR4 expression (cytoplasmic and nuclear) with clinicopathological parameters in cohort n_1_. Bold *p*-values are statistically significant.

n_1_: ccRCC Cohort	T-Stage (T > T2)	N-Stage (N+)	Lymphovascular Invasion (L+)	Vascular Invasion (V+)	Grading (G > G2)	Sarcomatoid Differentiation	Necrosis
Positive cytoplasmic CXCR4 expression	*p* = 0.5809	*p* = 0.2370	***p* = 0.0097**	*p* = 0.1298	*p* = 0.6861	*p* = 0.6142	*p* = 0.0634
Positive nuclear CXCR4 expression	*p* = 0.2204	*p* = 0.7859	*p* = 0.2411	*p* = 0.5148	*p* = 0.8787	*p* = 0.4132	*p* = 0.9616

**Table 5 ijms-24-05227-t005:** Univariate and multivariate Cox regression analysis for cancer-specific survival (CSS) and overall survival (OS). Bold *p*-values are statistically significant.

	Cancer-Specific Survival (CSS)						Overall Survival (OS)					
Covariate	Univariate analysis			Multivariate analysis			Univariate analysis			Multivariate analysis		
	*p*	HR	95% CI	*p*	HR	95% CI	*p*	HR	95% CI	*p*	HR	95% CI
Clinical Stage ≥ T2	**0.0093**	7.6407	1.6629 to 35.1085	0.1386	4.3516	0.6278 to 30.1617	**0.0046**	6.3617	1.7835–22.6920	0.1071	3.9410	0.7497 to 20.7186
Sarcomatoid features	**0.0003**	13.9040	3.4158 to 56.5963	0.8367	0.7732	0.0678 to 8.8149	**0.0014**	9.0238	2.3498–34.6527	0.8128	1.3192	0.1348 to 12.9154
N+	**<0.0001**	15.5824	4.6962 to 51.7044	0.1364	6.3868	0.5636 to 72.3807	**<0.0001**	9.1089	3.1287–26.5202	0.4155	2.3116	0.3107 to 17.1968
Venous invasion V+	**0.0092**	3.2142	1.3412 to 7.70	0.7326	0.7371	0.1292 to 4.2035	**0.0152**	2.6846	1.2140–5.9367	0.5763	0.6658	0.1610 to 2.7536
Lymphovascular invasion L+	**<0.0001**	35.9364	7.0777 to 182.4646	0.5824	2.1766	0.1380 to 34.3246	**<0.0001**	19.0835	4.4745–81.3896	0.4900	2.8076	0.1520 to 51.8496
Fuhrman grading G > 2	**0.0004**	8.9889	2.6983 to 29.9454	0.4745	2.0447	0.2909 to 14.3739	**0.0026**	5.5545	1.8337–16.8248	0.6026	1.6717	0.2440 to 11.4528
Cytoplasmic expression CXCR4	**0.0317**	4.4826	1.1483 to 17.4993	0.1386	4.3516	0.6278 to 30.1617	**0.0457**	3.0454	1.0272–9.0290	0.5233	1.6381	0.3626 to 7.4002
Nuclear expression CXCR4	0.5092	1.5775	0.4103 to 6.0648				0.5529	1.8713	0.2388–14.6646			

## Data Availability

The data presented in this study are available on request from the corresponding author. The data are not publicly available due to ethical restrictions (Institutional review board statement).

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
