# Peer review of "Differential Expression and Clinical Relevance of C-X-C Motif Chemokine Receptor 4 (CXCR4) in Renal Cell Carcinomas, Benign Renal Tumors, and Metastases"

_ijms, 2023, doi:10.3390/ijms24065227_

Round 1

Reviewer 1 Report

The authors conducted the CXCR4 protein evaluation by immunohistochemistry in renal cell carcinoma tissues and normal and benign renal tissues. However, major concerns must be addressed for publication.

The authors presented three collectives in Tables 1, 2, and 3. All these tables need formation to facilitate the reading. In table 1, the authors might remove T3c, and T4 lines since no samples from these T stages were included. In the same table, the authors reported R-Status, L-Statis and V-Status. It is not clear what these variables represent. Please clarify. Concerning Grading, please explain if the Fruhman grade was used.

Additionally, the authors reported the mean and 95% CI. Please explain if 95% CI was used or SD. Concerning Table 2, it should follow the same formatting as Table 1. The same is valid for Table 3. The metastatic site might be presented in decreasing numbers, being the most common site presented first.

Moreover, section 2.1, Figure 1, is not very informative. For this reason, it might be presented as supplementary material. In this figure, the n excluded samples must be displayed. Furthermore, in Figures 2 to 5, boxplots might be replaced by dot-plot with the median to represent better the distribution of staining pattern for the different comparisons. In line 138, the authors presented the median with 95% CI. However, after graph reading, the values shown seem to represent P25 and P75. Please clarify. In line 144, the authors wrote, "...in cyst (90), followed...". Please clarify if 90 is the median value. In line 150, the authors performed comparisons between metastatic sites. However, for several metastatic sites, only one sample was used. The authors might be more careful when analyzing and reporting these results.

In figure 3, the count reported in the tables is not in accordance with figure 1. Please clarify. In figure 5, it is not clear what tissues were used.

When reporting decimal numbers, the authors might replace commas with dots.

In section 2.3, the authors might clarify the 95% CI presented in line 197. In line 204, the authors reported that nuclear expression was not associated with survival. Data not shown might be presented as supplementary material.

Figure 6 is missing. Concerning Table 6, all the variables lost significance in the multivariate analysis, including classic prognostic factors such as the clinical stage. The authors might address this issue more in the discussion. Moreover, some categories were considered together, e.g. T2, 3 and 4 were considered together. Please explain why. In line 236, the authors reported Figure 7. Please clarify.

The authors might improve the discussion by addressing possible explanations for non-significant results.

Furthermore, in section TMA & immunohistochemistry, it is unclear how many colours were used per case. The immunochemistry protocol might be more detailed, namely, which antigen retrieval was used.

Figure 8 is reported twice.

Author Response

Reply to the Reviewers:

Reviewer #1:

Dear Reviewer #1,

Thank you for your review of our manuscript and the appreciated suggestions for its improvement. In the following, we will present point by point how we changed our manuscript according to your recommendations (these are in bold).

The authors presented three collectives in Tables 1, 2, and 3. All these tables need formation to facilitate the reading. In table 1, the authors might remove T3c, and T4 lines since no samples from these T stages were included. In the same table, the authors reported R-Status, L-Statis and V-Status. It is not clear what these variables represent. Please clarify. Concerning Grading, please explain if the Fruhman grade was used.

We have formatted Tables 1, 2, and 3 according to the reviewer's suggestions. In Table 1, T3c and T4 have been removed. We added the definition of the abbreviations R-status, L-status, and V-status in the footnote below the table and in the M&M section 4.1. In terms of grading, we supplemented that we used the Fuhrman grading.

Additionally, the authors reported the mean and 95% CI. Please explain if 95% CI was used or SD. Concerning Table 2, it should follow the same formatting as Table 1. The same is valid for Table 3. The metastatic site might be presented in decreasing numbers, being the most common site presented first.

We confirm, that reported values in table 1 for follow-up time are mean and 95%CI of the mean. The reformatted Table 2 is now structured like Table 1 including TNM classification; for this purpose, all radiological and pathological reports were analyzed, resulting in a significant expansion of Table 2. Table 3 was rearranged according to the recommendation, presenting the metastasis sites in descending frequency. 

Moreover, section 2.1, Figure 1, is not very informative. For this reason, it might be presented as supplementary material. In this figure, the n excluded samples must be displayed. Furthermore, in Figures 2 to 5, boxplots might be replaced by dot-plot with the median to represent better the distribution of staining pattern for the different comparisons. In line 138, the authors presented the median with 95% CI. However, after graph reading, the values shown seem to represent P25 and P75. Please clarify. In line 144, the authors wrote, "...in cyst (90), followed...". Please clarify if 90 is the median value. In line 150, the authors performed comparisons between metastatic sites. However, for several metastatic sites, only one sample was used. The authors might be more careful when analyzing and reporting these results.

Figure 1 was revised and converted into three supplementary figures to address the issue of excluded samples and evaluated samples (see detailed in the answer to the next comment). We replaced the Figures 2-5 and integrated the dots in the boxplots for a better representation of the distribution of staining pattern.

We rechecked our calculations and can confirm that the mentioned values in the text are the 95% CI. We further clarified in the text, that 90 is the median value for cysts (line 160). We have added the restriction of the described results as well as their analysis in more detail than before in the discussion section (line 397 - 399). 

In figure 3, the count reported in the tables is not in accordance with figure 1. Please clarify. In figure 5, it is not clear what tissues were used.

We unintentional used an older version of Figure 1. Figure 1 was now revised and made into 3 supplementary figures. This aims to help the reader understand the complex composition and evaluation of the three different collectives. During the revision process, we checked that the counts of the individual figures are correct and consistent. For figure 5 we added the missing information in the footnote below the figure and section 4.1.

When reporting decimal numbers, the authors might replace commas with dots.

We have changed the separation of decimal numbers from commas to dots within the entire document (including graphs).

In section 2.3, the authors might clarify the 95% CI presented in line 197. In line 204, the authors reported that nuclear expression was not associated with survival. Data not shown might be presented as supplementary material.

We revised the section and ensured that the 95% CI are now located closer to the corresponding means. The KM curves of the OS and CSS related to nuclear expression are now presented as supplementary figure 4.

Figure 6 is missing. Concerning Table 6, all the variables lost significance in the multivariate analysis, including classic prognostic factors such as the clinical stage. The authors might address this issue more in the discussion. Moreover, some categories were considered together, e.g. T2, 3 and 4 were considered together. Please explain why. In line 236, the authors reported Figure 7. Please clarify. The authors might improve the discussion by addressing possible explanations for non-significant results.

We initially used the description Figure 8 twice incorrectly, one of which was planned as Figure 6. The revision of all Figures has resulted in new labels. These can be found as an overview under the last comment of the reviewer. The mentioned loss of significance for all variables including validated prognostic factors is now addressed in more detail within the discussion section and underlined as potential limitation of the study (line 386-391). The grouping of categories for dichotomized evaluation was done in consideration of the overall small sample size to achieve a balanced statistical power, for example, between the high number of T1 tumors and higher-grade tumors.

Furthermore, in section TMA & immunohistochemistry, it is unclear how many colours were used per case. The immunochemistry protocol might be more detailed, namely, which antigen retrieval was used.

The TMA & immunohistochemistry section has been extensively revised (line 462 -479).

Figure 8 is reported twice.

The Figure 8 label was incorrectly given twice. The revision of the Figures according to the recommendations now results in the following labels:

Former Figure number

Description

New Figure Number

Figure 1

Distribution of evaluated specimens within the different collectives.

Supplementary Figures 1-3

Figure 2

Cytoplasmic and nuclear CXCR4 Expression for collective n1.

Figure 1

Figure 3

Cytoplasmic and nuclear CXCR4 Expression for collective n2.

Figure 2

Figure 4

Cytoplasmic and nuclear CXCR4 Expression for collective n3.

Figure 3

Figure 5

Cytoplasmic and nuclear CXCR4 Expression for combined samples of collective n1 and n3.

Figure 4

Figure 6 (incorrectly referred to as Figure 8)

KM-Analysis for cytoplasmic cxcr4 staining collective n1

Figure 5

Figure 7 and figure 8

Illustration of staining intensities for CXCR4 in malignant tissue (cytoplasmic, nuclear) and in benign tissue.

Figure 6

KM-Analysis for nuclear CXCR4 staining collective n1

Supplementary Figure 4 *New*

Reviewer 2 Report

This is an interesting study, is another work on this subject, which can be considered in the meta-analysis in this topic.

Some critical issue should be solved

One of the major criticism of this work concern the choice of the authors to put in the group  n.2, as benign lesions, oncocytoma, cyst and angiomyolipoma. Oncocytoma is an epithelial benign tumor but angiomyolipoma is a mesenchymal tumor and cystic lesions are not tumor so I think that neither angiomyolipomas or cysts should be included in this group.

The authors should explain why they didn’t consider all CCRCC together (group n.1 and n.2)  in the statistical analysis, they would have a larger group.

Another issue is that they considered metastatic lesions without their primary neoplastic lesion to compare the expression of CXCR4. The authors presented this as a limit of the study but could be interesting almost in a part of the cases.

The authors should update the TNM and grading if possible, on the base of ISIP/WHO for Clear el renal cell carcinoma and papillary renal cell carcinoma.

Minor issues:

In expression of CXCR4 row 130-131 page 5 the authors should modified benign kidney lesions with  normal kidney tissue because this part is referred to n.1 cohort. The Authors could insert the image of expression of CXCR4 in normal tissue in the panel of figure.

In the figure 1 the label under the cohort n. 2 is the same of cohort n.3  but why the authors used the term unknown primary tumor, the authors should explain it in the text or in the legend of figure.  In fact for 4 cases of the kidney tumor that the authors didn’t consider in group n2 could be either renal cell carcinoma NOS (not distinct subtype) or rare subtype.

The authors better explain in the text  their statements  in page 5 row 144-149.

The authors (pages 5-6)  describe their results comparing CXCR4 staining between  benign tissue, ccRCC and metastasis but they should be indicate if  they compare only ccRCC metastasis or metastasis of all renal cell carcinoma   regardless of type. It would be better compare metastasis of clear cell carcinoma only.

 In discussion (page 10 row 218) the authors statement is that nuclear expression of ccRCC is  low intense but the comparison is not the same . In n1 is true between  ccRCC and normal kidney tissue but for  n2 group they stated that ‘ there was no discernible…’ see page 5 row 146. So the authors should better explain it.

The authors could discuss their result in n1 where normal tissue have CXCR4 expression higher than ccRCC because in the literature this is not always true.

The authors found  a positive correlation between expression of cxcr4 and  lymphovascular invasion, this

findings is not very common in kidney tumor, the same result was described by Rasti ( quoted by the authors) so they could discuss it.

In the materials and methods:

 When the authors listed pathological prognostic  factor (page 12 row 332) should add necrosis because this parameter is present in the table.

The authors could add the mean number of the core for each case they carried out.

Author Response

Reviewer #2:

This is an interesting study, is another work on this subject, which can be considered in the meta-analysis in this topic.

Thank you for your review of our manuscript and the appreciated suggestions for its improvement. In the following, we will present point by point how we changed our manuscript according to your recommendations (these are in bold).

Some critical issue should be solved

One of the major criticism of this work concern the choice of the authors to put in the group  n.2, as benign lesions, oncocytoma, cyst and angiomyolipoma. Oncocytoma is an epithelial benign tumor but angiomyolipoma is a mesenchymal tumor and cystic lesions are not tumor so I think that neither angiomyolipomas or cysts should be included in this group.

The different histologic genesis of the benign lesions in the n2 collective is a valid and appreciated concern. However, we believe that there has been a misunderstanding here, presumably based on the incorrect use of the expression ‚benign kidney lesions‘ in line 144.

The heterogeneous collective n2, consisting of renal cell malignancies ccRCC, pap. RCC, chrom. RCC, and the benign lesions oncocytoma, angiomyolipoma, and cysts was not analyzed in a pooled way in any of our analyses. These subtypes were always analyzed within their respective subtype, e.g. all oncocytomas vs. all angiomyolipomas.

The comparison of ccRCC versus benign tissue (Fig 1) was performed only within the n1 collective. TMAs from benign lesions of collective n2 were not used. The benign normal kidney tissue used for the comparison in figure 1 comes only from collective n1 and had previously been pathologically identified as benign kidney tissue.

To address this potential for misunderstanding, we have replaced the word ‘lesion’ with ‘tissue’ in line 144 and further revised the M&M section. Footnotes of respective figures were revised to clarify.

The authors should explain why they didn’t consider all CCRCC together (group n.1 and n.2)  in the statistical analysis, they would have a larger group.

Collectives n1 and n2 serve different clinical questions in our study. Collective n1 is a highly detailed collective in its clinical, pathological and, above all, survival data. Combining the collectives would have resulted in an increase in the total number of samples at the expense of the evaluability of the clinical data, which is why we decided to maintain the segregation after weighing the advantages and disadvantages of such a combined evaluation in view of the still high number of ccRCC in collective n2. 

Another issue is that they considered metastatic lesions without their primary neoplastic lesion to compare the expression of CXCR4. The authors presented this as a limit of the study but could be interesting almost in a part of the cases.

We share the reviewer's opinion that the quality of the data would increase by a matched analysis between primary tumor tissue and corresponding metastatic tissue. Unfortunately, we cannot offer this for our collective n3: The vast majority of patients in this collective were referred to our university center for metastatic surgery only, so the primary tumors are not available for analysis. The remaining number of patients for whom both tissue types are available is so small that a grouped evaluation has no statistical value. To emphasize this desirable point, we have worded the relevant section of the text more clearly.

The authors should update the TNM and grading if possible, on the base of ISIP/WHO for Clear el renal cell carcinoma and papillary renal cell carcinoma.

Unfortunately, we cannot realize this request. The reason for this is the retrospective assembly of our collectives using the existing pathological reports in our hospital information system. Within our hospital, these reports are provided using the TNM classification. A pathologic re-evaluation of all patients to report them by ISUP/WHO classification is beyond the scope of our work presented here. 

To meet the reviewer request we have decided to collect the clinical and radiological data also for the collective n2 (with the included pap. and chrom. RCCs) and can offer a revised table n2. Furthermore, we would like to acknowledge the beneficial development for the assessment of RCCs by ISUP/ WHO classification and have therefore added a corresponding section within the limitations of the paper (line 399-415). 

Minor issues:

In expression of CXCR4 row 130-131 page 5 the authors should modified benign kidney lesions with  normal kidney tissue because this part is referred to n.1 cohort. The Authors could insert the image of expression of CXCR4 in normal tissue in the panel of figure.

We changed the word ‘lesions’ to ‘tissue’ (line 144). We have added an example image for staining of CXCR4 in benign tissue, although the applied evaluation criteria is the same as for malignant tissue.

In the figure 1 the label under the cohort n. 2 is the same of cohort n.3  but why the authors used the term unknown primary tumor, the authors should explain it in the text or in the legend of figure.  In fact for 4 cases of the kidney tumor that the authors didn’t consider in group n2 could be either renal cell carcinoma NOS (not distinct subtype) or rare subtype.

Figure 1 has now been moved to the supplementary section based on the recommendation of Reviewer #1. In order to show more precisely how the samples analyzed from the initial collectives were composed, we have revised figure 1 and made it into Supplementary Figure 1-3. This is to acknowledge the reviewer's commentary and to better reflect the individual collectives.

The authors better explain in the text  their statements  in page 5 row 144-149.

We have revised the corresponding section and added a reference to the post-hoc analysis (line 162-166).

The authors (pages 5-6)  describe their results comparing CXCR4 staining between  benign tissue, ccRCC and metastasis but they should be indicate if  they compare only ccRCC metastasis or metastasis of all renal cell carcinoma   regardless of type. It would be better compare metastasis of clear cell carcinoma only.

To clarify that only the metastatic tissue of ccRCC was analyzed here, we have added this to the text (line 173). In addition, the setup has been explained within the footnote of Figure 4 and in the M&M section.

 In discussion (page 10 row 218) the authors statement is that nuclear expression of ccRCC is  low intense but the comparison is not the same . In n1 is true between  ccRCC and normal kidney tissue but for  n2 group they stated that ‘ there was no discernible…’ see page 5 row 146. So the authors should better explain it.

To clarify the different collectives compared, we have revised the relevant paragraph within the discussion (line 434).

The authors could discuss their result in n1 where normal tissue have CXCR4 expression higher than ccRCC because in the literature this is not always true.

We added a section to discuss this between the lines xx and xx.

The authors found  a positive correlation between expression of cxcr4 and  lymphovascular invasion, this findings is not very common in kidney tumor, the same result was described by Rasti ( quoted by the authors) so they could discuss it.

We added a short passage to address this comment within our discussion section (line 282 – 288).

In the materials and methods:

 When the authors listed pathological prognostic  factor (page 12 row 332) should add necrosis because this parameter is present in the table.

We added necrosis to this section.

The authors could add the mean number of the core for each case they carried out.

We revised Figure 1 to supplementary figures 1-3 to explain how the patients of the collectives reflects in analyzed cores.

Round 2

Reviewer 1 Report

Concerning tables, caption letter should be use for gender.
The authors should use the word "cohort". Cohort is related with patients. When the authors are talking about samples, might be use the word "serie".

When significant, the auhtor should report the exat p-value (except if p<0.001). A table with these value might be included in the supplementary data.

In results, the authors reported "Nuclear expres sion in primary RCC differed significantly from benign tissue and metastasis, while no  significant difference between benign tissue and metastases were noted (Figure 4). Regression analysis revealed no significant correlation of cytoplasmic and nuclear staining scores (p=0.47)". However, figure 4 shows differences between benign and metastatic tissues in both stainings (nuclear and cytoplasmatic). Please clarify.
The discussion might be finished with the major findings of the study rather the limitations.

Concerning methods, please report how many cores was used per sample for TMA construction. Please report if antigen retrieval was used for immunohistochemistry. 

In statistical analysis, the author must perform only non-parametric test for comparasions among groups. T-test must be replaced for Mann-U-Whitney test. Pos-hoc correction must be applied.

Quality of suplemmentary figures might be increased.
Graphs presented are not very appealing to the reader. 

Author Response

Reply to the Reviewers:

Dear Reviewer,

Thank you for your reasoned comments and suggestions. Based on these, we revised our manuscript. The resulting changes are added after the corresponding comment.

Concerning tables, caption letter should be use for gender.

Caption letters were used for gender.

The authors should use the word "cohort". Cohort is related with patients. When the authors are talking about samples, might be use the word "serie".

We have been replacing ‘collective’ with the word ‘cohort’ throughout the manuscript, corresponding tables, and figures.

When significant, the auhtor should report the exat p-value (except if p<0.001). A table with these value might be included in the supplementary data.

We reviewed the text to ensure that all p-values were placed in the correct location and filled in the missing p-values (e.g. lines 166, 170). For pairwise comparisons of subgroups MedCalc performs a test according to Conover, 1999.The desired significance level for the post-hoc test is prespecified as <0.05. If the Kruskal-Wallis test results in a p-value less than this predefined significance level, the output is referring to the prespecification and gives only the feedback, whether this value is significant or not. We kindly request to accept this representation.

In results, the authors reported "Nuclear expression in primary RCC differed significantly from benign tissue and metastasis, while no  significant difference between benign tissue and metastases were noted (Figure 4). Regression analysis revealed no significant correlation of cytoplasmic and nuclear staining scores (p=0.47)". However, figure 4 shows differences between benign and metastatic tissues in both stainings (nuclear and cytoplasmatic). Please clarify.
The discussion might be finished with the major findings of the study rather the limitations.

We thank the reviewer for pointing out this mistake in the manuscript. The section has been revised and corrected (lines 165 and 169-171).

Concerning methods, please report how many cores was used per sample for TMA construction. Please report if antigen retrieval was used for immunohistochemistry.

We added the information, that 2 cores were taken for all kind of tumor tissue and one core for benign kidney tissue (lines 408-411).

Antigen retrieval was applied. We have added this in the section of the IHC (line 412).

In statistical analysis, the author must perform only non-parametric test for comparasions among groups. T-test must be replaced for Mann-U-Whitney test. Pos-hoc correction must be applied.

We replaced the t-Test with the Mann-U-Whitney test, p-value of nuclear staining for collective n1 was changed accordingly (line 450 and line 141, figure 1). When comparing cytoplasmic staining, the p-value did not change by applying the Mann-Whitney U test (p for both tests <0.0001).

Quality of suplemmentary figures might be increased.
Graphs presented are not very appealing to the reader.

All Figures have been edited and improved in quality to be more appealing to the reader